# Experimental–Density Functional Theory (DFT) Study of the Inhibitory Effect of Furan Residues in the Ziegler–Natta Catalyst during Polypropylene Synthesis

**DOI:** 10.3390/ijms241814368

**Published:** 2023-09-21

**Authors:** Joaquín Hernández-Fernández, Esneyder Puello-Polo, Edgar Marquez

**Affiliations:** 1Chemistry Program, Department of Natural and Exact Sciences, San Pablo Campus, University of Cartagena, Cartagena 130015, Colombia; 2Chemical Engineering Program, School of Engineering, Universidad Tecnológica de Bolivar, Parque Industrial y Tecnológico Carlos Vélez Pombo, Cartagena 130001, Colombia; 3Department of Natural and Exact Science, Universidad de la Costa, Barranquilla 080002, Colombia; 4Group de Investigación en Oxi/Hidrotratamiento Catalítico Y Nuevos Materiales, Programa de Química-Ciencias Básicas, Universidad del Atlántico, Puerto Colombia 081001, Colombia; esneyderpuello@mail.uniatlantico.edu.co; 5Grupo de Investigaciones en Química Y Biología, Departamento de Química Y Biología, Facultad de Ciencias Básicas, Universidad del Norte, Barranquilla 081007, Colombia

**Keywords:** furan, Ziegler–Natta catalyst, polypropylene synthesis, flow rate, density functional theory (DFT), mechanical properties, adsorption affinity

## Abstract

In this experimental–theoretical study, the effect of furan on Ziegler–Natta catalyst productivity, melt flow index (MFI), and mechanical properties of polypropylene were investigated. Through the analysis of the global and local reactivity of the reagents, it was determined that the furan acts as an electron donor. In contrast, the titanium of the ZN catalyst acts as an electron acceptor. It is postulated that this difference in reactivity could lead to forming a furan–titanium complex, which blocks the catalyst’s active sites and reduces its efficiency for propylene polymerization. Theoretical results showed a high adsorption affinity of furan to the active site of the Ti catalyst, indicating that furan tends to bind strongly to the catalyst, thus blocking the active sites and decreasing the availability for propylene polymerization. The experimental data revealed that the presence of furan significantly reduced the productivity of the ZN catalyst by 10, 20, and 41% for concentrations of 6, 12.23, and 25.03 ppm furan, respectively. In addition, a proportional relationship was observed between the furan concentration and the MFI melt index of the polymer, where the higher the furan concentration, the higher the MFI. Likewise, the presence of furan negatively affected the mechanical properties of polypropylene, especially the impact Izod value, with percentage decreases of 9, 18, and 22% for concentrations of 6, 12.23, and 25.03 ppm furan, respectively.

## 1. Introduction

In the search for solutions to mitigate the environmental impact caused by polymers from their production to their management as waste, in recent years, attempts have been made to obtain raw materials of natural origin and recycle them to promote a green economy. However, it is worrying that of the more than 369 million tons produced annually, only 9% of the polymers generated are recycled in the United States and 12% globally [1]. In the particular case of polypropylene (PP), the post-consumer recycling rate worldwide is approximately 1%; this worsens when the PP obtained does not meet the physicochemical parameters, which usually occurs when the Ziegler–Natta (ZN) catalyst used for its synthesis suffers from poisoning by impurities; these products must be discarded, contributing to large amounts of PP ending in landfills, aggravating plastic pollution [2,3,4]. To address this environmental problem, pyrolysis has emerged as a recycling alternative. This chemical and thermochemical process involves the thermal decomposition of the polymeric material in the absence of oxygen or with a limited amount of it. During pyrolysis, polymers are subjected to high temperatures in a controlled environment, causing them to break down into simpler components such as gases, liquids, and charred solid residues. In the case of PP, pyrolysis can be used to obtain constituent monomers from discarded PP, thus closing the life cycle of materials and reducing dependence on fossil resources. In addition, during pyrolysis, substances such as hydrocarbons and furans are generated, which can be used as raw materials for the manufacture of other industrial products and to generate energy [5,6].

Furan C_4_H_4_O is a heterocyclic organic compound with a five-membered ring composed of four carbon atoms and one oxygen atom in its structure. It is a colorless, lipophilic, volatile, and highly flammable liquid. Industrially, furan is essential due to its wide range of applications. It is a valuable intermediate in the production of resins, lacquers, pesticides, and drugs, and it also plays a vital role in synthesizing various chemical compounds, such as tetrahydrofuran, pyrrole, and thiophene. It even can act as a diene in Diels–Alder reactions. In these reactions, furan can combine with dienophiles, compounds that contain double bonds, resulting in the efficient and selective formation of complex cyclic compounds. While it is a valuable compound, it has garnered significant attention because it also forms naturally as a byproduct in the thermal processing of foods. It has been detected in various food products and beverages, such as coffee, chips, and canned foods, which has raised concerns about its effects on human health. Organizations such as the International Agency for Research on Cancer (IARC) have classified it as a possible 2B carcinogen, and the United States Food and Drug Administration (FDA) has declared it a pathogen if found in food [7,8,9,10,11,12].

When recycled raw materials are used to produce polypropylene (PP), which contains traces of furan, there is a risk of generating nonproductive catalytic complexes that affect the speed and selectivity of the reaction, altering the isotactic structure of the resulting PP. Experimental studies [13], have shown productivity losses of the ZN catalyst of 10%, 20%, and 41% for the copolymers synthesized with ethylene enriched with 6, 12, and 25 ppm furan, respectively. Furthermore, as the furan concentration increases, a significant decrease is observed in the melt flow index (MFI) and PP’s thermal (TGA) and mechanical properties. These results indicate the detrimental effect of furan on the quality and yield of PP produced by the ZN catalyst. It is essential to address this problem to improve the efficiency of PP production and promote more sustainable practices in the polymer industry [13].

Unfortunately, no computational theoretical investigations address furan poisoning to the Ziegler–Natta (ZN) catalyst. This limitation highlights the importance of further theoretical support to comprehensively understand the inhibition caused by this compound. By delving into the analysis of heterocyclic compounds and evaluating their chemical properties, reactivity, and global and local descriptors, we can obtain a complete view of how they interact with the ZN catalyst and how this affects the polymerization process. In this study, we propose an innovative approach based on density functional theory (DFT) and Gaussian 09 software to investigate the furan–titanium interaction in detail. Through the application of global and local descriptors, we analyze how the presence of furan impacts the reactivity of the catalyst and, consequently, influences the quality of the resulting polymer. In addition, it is essential to understand the extent of the damage caused to the polymer and to analyze its sustainability to promote more responsible practices, thus reducing material waste and contributing to a more sustainable and environmentally friendly approach. Our research seeks to generate valuable knowledge that supports scientific research and the production of polypropylene more ecologically and in harmony with our environment.

## 2. Results and Discussion

First, the global reactivity indices of the reagents were calculated to determine the demand for electrons (normal or inverse) and the polarity (polar or nonpolar character). A general interpretation was also made on the effect of substituents on the electrophilicity of furan. Second, local reactivity descriptors were calculated to rationalize the interaction between furan and titanium tetrachloride (TiCl_4_).

### 2.1. Character of the Reaction

We consider the D–A reactions between furan and TiCl_4_. In a D–A reaction, the donor (furan) contributes electrons, and the acceptor (TiCl_4_) accepts them. TiCl_4_ contains two active sites on the titanium atom, which acts as an electron acceptor, and furan includes an oxygen atom, which can act as an electron donor. Therefore, these D–A reactions are expected to proceed with normal electron demand (NED), which means that the donor donates electrons to the acceptor naturally to form the product. In the average demand for electrons, the most critical interactions occur between the highest occupied molecular orbitals (HOMO) of the donor and the lowest unoccupied molecular orbitals (LUMO) of the acceptor (see Figure 1). In this case, the HOMO of the inhibitor (furan), which contains the oxygen atom with a lone pair of electrons, is expected to interact with the LUMO of TiCl_4_, which is the orbital available to receive the donor electrons.

The reactions in question have a specific demand for electrons, which can be predicted by an analysis based on density functional theory (DFT) that considers the general properties of the reactive molecules. Table 1 summarizes the essential characteristics of the relevant reagents to this analysis. The chemical potential represents the charge transfer capacity of the system in its ground state geometry. In this case, the negative chemical potential of furan indicates that it tends to donate electrons. In contrast, the higher negative chemical potential of TiCl_4_ suggests that it has a more remarkable ability to accept electrons. Furan has a higher HOMO energy than TiCl_4_, which implies that furan is a better electron donor. The difference in HOMO energies between furan and TiCl_4_ (N) indicates a tendency toward electron transfer from furan to TiCl_4_, favoring the formation of a donor–acceptor complex between both molecules. Finally, the difference in electronegativities (χ) suggests that TiCl_4_ has a greater capacity to accept electrons than furan, which is consistent with its role as an electron acceptor in the D–A reaction.

The parameter ω (omega) is used to evaluate the global electrophilic character of a molecule. A low value of ω indicates that the molecule has a more electrophilic character, while a high value of ω indicates a more nucleophilic character. Marginal electrophiles have values of ω closer to zero but still positive. In the reactivity analysis, if ω ≈ 0 or ω is slightly positive, the molecule has a somewhat electrophilic capacity, meaning it can act as a marginal electrophile in specific reactions. The value closest to zero is that of furan (1.1656 eV), indicating that furan has marginal electrophilicity compared to TiCl_4_. Therefore, in the DA reaction between furan and TiCl_4_, furan acts as the marginal electrophile, the electron donor in this interaction. TiCl_4_, on the other hand, would serve as the electron acceptor in this reaction.

### 2.2. Analysis of Condensed Fukui Functions and Their Relevance in the Inhibition of Furan to the Z–N Catalyst (TiCl_4_)

In this study, an exhaustive analysis of the condensed Fukui functions for the furan atoms and the Z–N catalyst (TiCl_4_) was carried out using the density functional theory (DFT) method and optimized structures using Gaussian09 based on B3LYP/6-311G. The main objective was to understand how furan could affect the reactivity of the Z–N catalyst and its possible involvement in the inhibition of polymer synthesis.

In the specific case of the furan compound, calculations were performed and Table 2 generated using the values of the Fukui functions (f^o^ and f^+^) for each site in the molecule. These results help us understand the chemical reactivity of furan and give us information about the reactivity and selectivity of specific sites within the molecule. This information is valuable in understanding how furan can interact and react with other chemical species in different contexts.

The results obtained for furan indicated that atoms 1, 2, 3, and 4 present a higher affinity to accept electrons (f^+^), while atom 5 shows a nucleophilic tendency to donate electrons (f^−^) (see Figure 2). This suggests that furan is a potentially nucleophilic electron donor and could interact with electrophilic species.

On the other hand, the results for the titanium atom in the Z–N catalyst revealed that atom 1 (Ti) shows a high affinity to accept electrons (f^+^), indicating that the catalyst is electrophilic.

These results suggest a difference in reactivity between furan and the Z–N catalyst, with furan showing a higher tendency to donate electrons (nucleophilic) and the titanium atom of the catalyst showing a higher affinity to accept electrons (electrophilic). This difference in reactivity could have essential implications in the synthesis of the family of polymers catalyzed by the Z–N catalyst. It is postulated that furan could inhibit the catalyst by donating electrons to the titanium atom, thus blocking the catalyst’s active sites and reducing the availability of sites for other reactive species, such as propylene.

### 2.3. Adsorption Energies

This section investigates how furan interacts with the titanium (Ti) active site on the MgCl_2_ surface. To carry out these calculations, a different model of the MgCl_2_ surface was used, which was obtained from the relaxed MgCl_2_ surfaces, as shown in Figure 3. This MgCl_2_ sheet has (110) (quadruple) surfaces on both sides. The choice of this model was based on previous calculations that suggested that the coordination of TiCl_4_ to the (104) plane is weak or even unstable. In comparison, the coordination of TiCl_4_ to the (110) plane is energetically favorable [14].

These results help us to understand how furan can specifically affect the Ti active site on the MgCl_2_ surface and how this interaction can influence the catalytic reaction being studied. To determine the energy of adsorption (*E_ad_*) of furan on TiCl_4_, we used Equation (1), substituting the values of *E_Mg/P_*, *E_Mg_*, and *E_p_* obtained through computational methods.
(1)$$E_{ad}=E_{Mg/P}-E_{Mg}-E_{P}\;$$
$$E_{ad}=\left(-18150.601746\right)-\left(-17920.557056\right)-\left(-230.009751\right)\;Hartree$$
$$E_{ad}=-0.034939795\;Hartree≅-21.924\;\mathrm{kcal}/\mathrm{mol}$$

The negative value of *E_ad_* indicates that the adsorption of furan on TiCl_4_ is exothermic; that is, it releases energy in the adsorption process. This suggests that the interaction between furan and the catalyst’s active site is favorable and promotes the formation of the adsorbed species. The high negativity of this value indicates that the Ti–furan complex is a suitable acceptor of electrons and is willing to accept electrons from other atoms or molecules.

The energy of adsorption (*E_ad_*), −21.924 kcal/mol, means that the adsorption of furan on the active site of the Ti catalyst is highly favorable from the thermodynamic point of view. Such good adsorption indicates that furan tends to bind strongly to the catalyst’s active site, which can occupy reaction sites and decrease the availability of active sites for propylene. In this context, if furan adsorption is so strong, it could act as an inhibitor of propylene insertion. By occupying the active sites and maintaining a strong interaction with the catalyst, furan can block or hinder the access of propylene to the reaction site, thus reducing the probability that the propylene insertion reaction will occur efficiently.

### 2.4. Effects of Furan on PP’s MFI, Productivity, and Mechanical Properties

Figure 4 shows a proportional relationship between the ppm furan and the MFI of PP1, PP2, and PP3. The MFI of the synthesized PP without furan was 20. PP with MFI of 20.77, 23.47, and 26.6 g/10 min were characterized when ppm furan was added to the polymerization process with 6, 12.23, and 25.03 ppm, respectively. This supports what is observed in Figure 4, which suggests that furan impairs the Ziegler–Natta catalytic activity and, therefore, the polymerization reaction. The oxygen atom in the furan ring reacts with the active center of the titanium of the Ziegler–Natta catalyst to generate a stable complex that prevents the growth of the PP chain [15,16,17]. A similar effect has been reported [18] when other chemical impurities are present during the PP synthesis.

Figure 4 also shows how the presence of furan affects the mechanical properties of the polymer. When analyzing the mechanical properties of the product obtained, a clear difference stands out between the polymer with concentrations of 0 ppm and the polymers with 6, 12.23, and 25.03 ppm. This suggests that the presence of furan directly impacts these properties: as the furan concentration increases, the impact values decrease.

The average Izod impact values for the polymer were 47, 42.47, 37.6, and 27.83 ft-Lb*in, for 0, 6, 12.23, and 25.03 ppm, respectively. The Izod impact trend was inversely proportional to the furan concentration, showing 9, 18, and 22% decreases. This variation in the data is mainly due to changes in the polymer structure due to the formation of new compounds with different functional groups. In addition, incomplete polymerization and the presence of the oxygen atom in furan also directly influence the mechanical properties of the polymer.

Regarding the productivity of the Ziegler–Natta catalyst, it is shown that it decreases as the furan concentration increases. When there is no furan presence, the ZN catalyst’s productivity is 47 MT/kg. However, when average concentrations of 6, 12, and 25 ppm furan are added to propylene, productivity is reduced by 10, 20, and 41%, respectively. In other words, furan hurts the efficiency of the Ziegler–Natta catalyst, and this reduction in productivity increases with higher amounts of furan present in the process.

### 2.5. Analysis of the Proposed Inhibition Mechanism for Furan

Figure 5 shows the proposed mechanism for inhibiting the Ziegler–Natta catalyst in the presence of furan. The first route is the union of the furan oxygen atom to an active site of the titanium of the catalyst through a coordination type bond. At the beginning of the inhibition process, the furan, which is a molecule that contains a five-membered ring with an oxygen atom, approaches the Ziegler–Natta catalyst. The oxygen atom of furan has a lone pair of electrons that can coordinate with the metal center of the catalyst, in this case, titanium. Furan binds to the active site of titanium through a coordinate bond, forming a furan–titanium complex. This binding blocks the catalyst’s active sites, decreasing its ability to activate olefin monomer molecules and initiate polymerization. In the second route, the union of ethyl occurs, which is released from the aluminum alkyl cocatalyst (TEAL). In the Ziegler–Natta catalyst polymerization process, an aluminum alkyl (TEAL) cocatalyst activates the titanium catalyst. However, when furan is present, it can compete with olefins for TEAL coordination sites. This can lead to an ethyl (alkyl group) breaking off the TEAL and attaching to the furan, forming a furan–ethyl complex. This additional binding of ethyl to furan further reduces the availability of active sites in the Ziegler–Natta catalyst for olefin polymerization, thus contributing to the inhibition of the polymerization process [13].

## 3. Materials and Methods

### 3.1. Reagents

This study used a state-of-the-art spherical Ziegler–Natta catalyst with MgCl_2_ support containing 3.6% Ti by weight. We used diisobutyl phthalate (DIBP) as an internal donor, with a purity of 99.99% from Sudchemie in Munich, Germany. As a cocatalyst, we used 98% pure triethylaluminum (TEAL) from Merck, Darmstadt, Germany, and 99.9% pure methylcyclohexyl dimethoxy silane. Lynde provided the hydrogen and nitrogen gases we use. Finally, the propylene used was obtained from Airgas with a purity of 99.999%.

### 3.2. Polypropylene Polymerization

To synthesize polypropylene, the methodology proposed in [15], which uses a Ziegler–Natta catalyst [19,20,21], was used. The process starts in a nitrogen-purged fluidized bed reactor. Then, hydrogen and propylene are introduced, which provide fluidization and absorb the heat of the reaction. Ziegler–Natta catalyst, TEAL, and an agent to control selectivity were incorporated, along with nitrogen. The synthesis was performed at 70 °C and 27 bar pressure in batch conditions [14,18,22,23,24,25,26,27,28,29,30,31,32,33,34,35]. The resin from the reactor was carried to a steam and hot nitrogen purge tower to remove hydrocarbons from the system.

### 3.3. Analysis by Gas Chromatography with Mass Selective Detector (GC-FID)

Furan quantification was performed using a gas chromatograph with a mass detector Agilent 7890B (Agilent Technologies, Inc., Santa Clara, CA, USA). For the injections of the samples, two points were used, one at the front with conditions (250 °C, 7.88 psi, 33 mL min^−1^) and one at the rear with (250 °C, 11.73 psi, 13 mL min^−1^). The chromatograph was programmed to warm gradually, starting at 40 °C for 3 min, then increasing to 60 °C in 4 min, and finally reaching 170 °C at a rate of 35 °C per minute. These adjustments allowed effective separation of the sample components, which ensured accurate measurements of the amount of furan.

### 3.4. Computational Methods

#### 3.4.1. Global Furan Reactivity Descriptors: Theoretical Basis

The frontier molecular orbital (FMO) theory is a tool that helps us understand how donor (D) and acceptor (A) molecules react. It has been widely used successfully to explain the reactivity and regiochemistry of these reactions [36]. However, certain limitations have been observed in its ability to predict regioselectivity in some cases [37]. For this reason, together with the FMO model, other reactivity indices based on the density functional theory (DFT) have also been used to evaluate the regiochemistry of the reaction under study. These DFT-based descriptors have proven more reliable in situations where FMO theory has not accurately predicted the reactivity and regioselectivity of specific D–A reactions. Recent studies have supported this approach and have validated that DFT descriptors provide more accurate results in some cases [38]. Recently, new concepts and parameters based on density functional theory (DFT) have been introduced that are valuable for modeling some substances’ chemical reactivity and selectivity. These parameters can be used as reactivity descriptors, either globally or locally. An example of these parameters is the chemical hardness (*η*), which describes the resistance of the chemical potential to change due to the number of electrons. Likewise, the electronic chemical potential (*μ*) is related to the charge transfer capacity of the system in its ground state geometry. Both magnitudes can be approximated as a function of the energies of the frontier molecular orbitals *HOMO* (higher occupied orbitals) and *LUMO* (lower unoccupied orbitals) using the following expressions (Equations (2) and (3)) [39].
(2)$$\eta =\left(\varepsilon_{LUMO}-\varepsilon_{HOMO}\right)$$
(3)$$\mu =\frac{\left(\varepsilon_{LUMO}+\varepsilon_{HOMO}\right)}{2}$$

The global electrophilicity index (*ω*), presented by Parr et al., is a valuable parameter in the study of chemical reactivity since it allows us to quantitatively classify the overall electrophilic character of a molecule on a unique relative scale [40]. The definition of this index is as follows:(4)$$\omega =\frac{\mu^{2}}{2\eta }$$

The difference in overall electrophilicity power between reagents provides valuable information about the polarity of D–A processes in a reaction [41]. This difference has been suggested as a measure of the polar character of the reaction.

#### 3.4.2. Furan Local Reactivity Descriptors: Uka Fukui

On the other hand, local reactivity indices are linked to the selectivity of the site in a chemical reaction. These parameters reflect the specific locations on a molecule where the pattern of reactivity established by the global quantities is expected to occur. An important example of a local reactivity parameter is the Fukui function, proposed by Parr et al. [40]. Subsequently, other local reactivity parameters have been introduced, such as softness [42], hardness [43], electrophilicity [44], and nucleophilicity [45]. These indices provide a more detailed and specific view of the reactivity at certain sites of the molecules involved in the reaction.

Equation (5) provides a simple and direct formalism to obtain the Fukui function from a relation with the frontier molecular orbitals (FMO) [46]. The condensed Fukui function for an electrophilic (nucleophilic) attack involves the HOMO (LUMO) FMO coefficients (c) and atomic overlap matrix elements (*S*).
(5)$$f_{k}{}^{\alpha }=\sum \mu \epsilon k{\left|C_{\mu \alpha }\right|}^{2}+\sum v\neq \mu \;C_{\mu \alpha }C_{v\alpha }S_{\mu v}$$

On the other hand, Equation (6) has been introduced to analyze which atomic site of a molecule’s maximum power of electrophilicity will be developed [41].
(6)$$\omega_{k}=\omega f_{k}{}^{+}$$

In addition, the first approaches toward a quantitative description of nucleophilicity through a regional reactivity index have also been reported. Equation (7), developed by Domingo et al., allows us to identify a molecule’s most nucleophilic site and evaluate the activation/deactivation caused by different substituents in the aromatic electrophilic substitution reactions of aromatic compounds [47].
(7)$$N_{k}=Nf_{k}{}^{-}$$

In Equation (8), $\epsilon_{HOMO,TCE}$ represents the *HOMO* energy of tetracyanoethylene (*TCE*), used as a reference molecule due to its low *HOMO* energy in a series of molecules previously considered. *N* is the global nucleophilicity index, and *N_k_* is its local counterpart.
(8)$$N=\left(\epsilon_{HOMO,Nu}-\epsilon_{HOMO,TCE}\right)$$

#### 3.4.3. Computational Details

Recent studies have shown that the B3LYP [48] method, even using the 6-31G(d) set of bases, is appropriate for modeling D–A reactions involving medium-sized molecules. This methodology has been successfully tested in various combinations of different compounds, giving satisfactory results. Therefore, for the species mentioned in this study, the geometries were obtained using a complete optimization with the B3LYP/6-311G(p,d) level using the GAUSSIAN16 program (Appendix A).

Chemical hardness, chemical potential, and global electrophilicity index were calculated using Equation (2).

To obtain the regional Fukui functions of furan, single-point calculations were performed on the optimized structure in the ground state using different levels of theory and basic ensembles (Equation (5)). A program has been developed and tested that reads the FMO coefficients and the overlap matrix from the Gaussian output files and performs the necessary calculation. Once the Fukui functions were obtained, the local values of electrophilicity and nucleophilicity were calculated (Equations (6) and (7)).

## 4. Conclusions

The experimental–theoretical study demonstrated that the presence of furan in the polymerization process significantly impacts the productivity of the Ziegler–Natta catalyst, the MFI melt index, and the mechanical properties of polypropylene. In Section 2.1, the theoretical analysis showed that furan acts as an electron donor (nucleophilic), while the titanium atom of the Ziegler–Natta catalyst acts as an electron acceptor (electrophilic). This difference in reactivity suggests that furan may interact with the catalyst, forming a complex that blocks the active sites and decreases the availability of sites for propylene polymerization. In Section 2.2, the theoretical data of the condensed Fukui functions revealed that furan has a higher tendency to donate electrons. At the same time, titanium from the Z–N catalyst shows a higher affinity to accept electrons. This difference in reactivity also supports the interaction between furan and catalyst, suggesting that furan could act as a polymerization inhibitor. In Section 2.3, it was found that the adsorption energy of furan on the active site of the Ti catalyst is highly favorable from the thermodynamic point of view. This strong adsorption indicates that the furan tends to bind firmly to the catalyst, blocking the active sites and reducing the probability that the propylene insertion reaction will occur efficiently. Finally, in Section 2.4, the experimental data showed that the presence of furan in the polymerization process decreased the productivity of the Ziegler–Natta catalyst significantly. As the furan concentration increases, productivity decreases by 10, 20, and 41% for 6, 12.23, and 25.03 ppm furan concentrations, respectively. In addition, the presence of furan affected the MFI melt index of the polymer, showing a proportional relationship between the furan concentration and the MFI: the higher the furan concentration, the higher the MFI of the polymer. It was also observed that the presence of furan affected the mechanical properties of the polymer, especially the impact Izod value. As the furan concentration increased, the impact values decreased, showing percentage decreases of 9, 18, and 22% for concentrations of 6, 12.23, and 25.03 ppm furan, respectively.

## Figures and Tables

**Figure 1 ijms-24-14368-f001:**
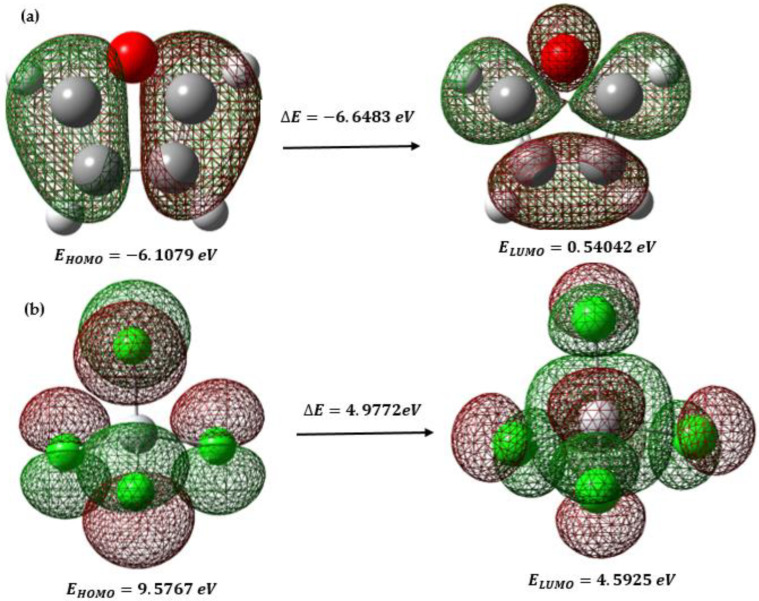
(**a**) HOMO–LUMO orbitals of furan; (**b**) HOMO–LUMO orbitals of TiCl_4_.

**Figure 2 ijms-24-14368-f002:**
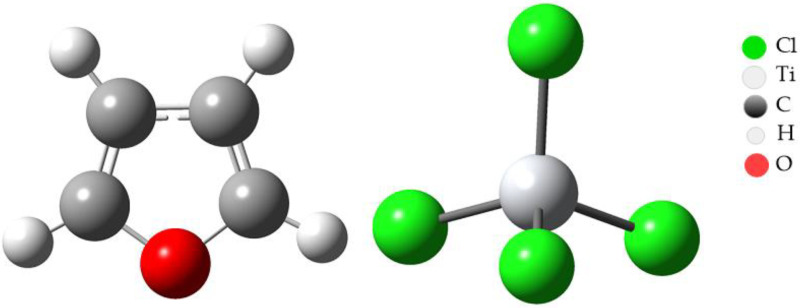
Optimized structure of furan and TiCl_4_.

**Figure 3 ijms-24-14368-f003:**
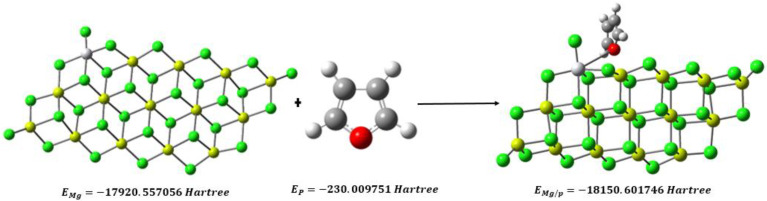
Furan adsorption on the (110) surface of the Ziegler–Natta catalyst.

**Figure 4 ijms-24-14368-f004:**
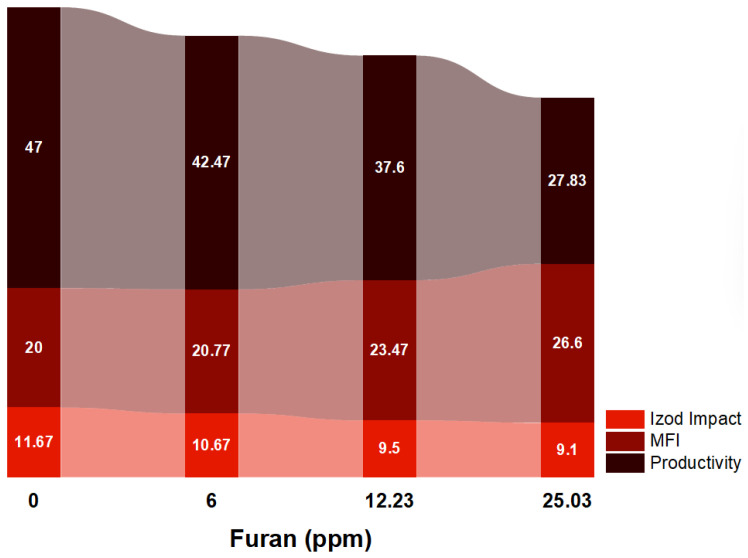
Analysis of productivity, MFI, and Izod impact of PP with traces of furan.

**Figure 5 ijms-24-14368-f005:**
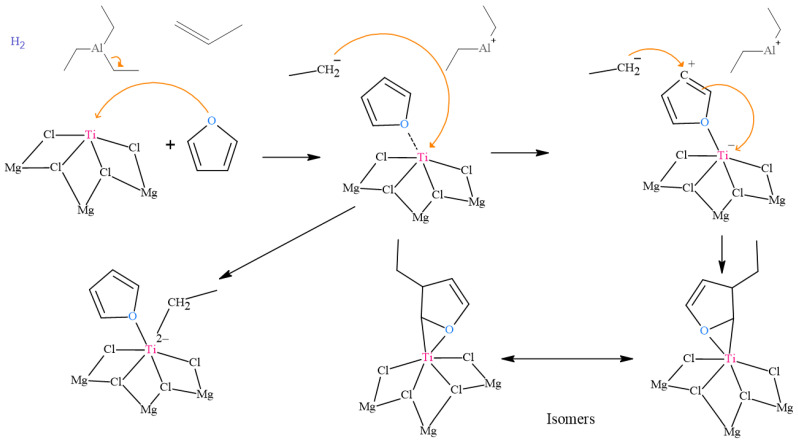
Proposed inhibition mechanism for furan.

**Table 1 ijms-24-14368-t001:** General characteristics of the reactive molecules. All values are expressed in electron volts (eV).

Properties	Furan	TiCl_4_
E_HOMO−1_	−7.4412	−9.5773
E_HOMO_	−6.1079	−9.5767
∆E	6.6483	4.9772
E_LUMO_	0.5404	−4.5925
E_LUMO+1_	2.1372	−4.5925
η	3.3241	2.4886
χ	2.7837	7.0846
S	0.3010	0.4018
µ	−2.7837	−7.0846
ω	1.1656	10.0843
N	3.0530	0.0992

**Table 2 ijms-24-14368-t002:** Local reactivity descriptors of furan and TiCl_4_.

Compound Number	Furan				TiCl_4_			
	f^+^	f^+^	f^o^	∆f	f^+^	f^+^	f^o^	Δf
1	0.3008	0.3668	0.3338	−0.066	0.7969	0.0001	0.3985	0.7969
2	0.0975	0.1329	0.1152	−0.0354	0.0507	0.3742	0.2124	−0.3235
3	0.0975	0.133	0.1153	−0.0354	0.0504	0.164	0.1072	−0.1136
4	0.3008	0.3669	0.3338	−0.066	0.0512	0.374	0.2126	−0.3228
5	0.2033	0.0005	0.1019	0.2028	0.0508	0.0877	0.0693	−0.0369

## Data Availability

The data presented in this study are available on request from the corresponding author.

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
