# Peer review of "Experimental–Density Functional Theory (DFT) Study of the Inhibitory Effect of Furan Residues in the Ziegler–Natta Catalyst during Polypropylene Synthesis"

_ijms, 2023, doi:10.3390/ijms241814368_

Round 1

Reviewer 1 Report

The current study titled “Experimental-density functional theory (DFT) study of the inhibitory effect of furan residues in the Ziegler-Natta catalyst during polypropylene synthesis” Ref: ijms-2592566, deals with an important subject. Minor revisions are needed.

- Figure 1 do not identify the (a) and (b) for the mentioned photos.

- All the obtained calculations due to DFT, FMO, …………… should be in the supplementary file.

- Table 1 should be revised. It is not well organized in the PDF version received (line numbers are overlapped with the first column).

Author Response

Dear

Thank you for evaluating this research.

We have made all the corrections you requested.

We have also included a document with supplementary information.

Kind Regards

Reviewer 2 Report

In this paper, the authors explored the effect of furan on Ziegler-Natta catalyst productivity, melt flow index (MFI), and mechanical properties of polypropylene were investigated. The authors proposed the presence of furan reduces the productivity of Z-N catalysts through simulation calculations and reduces the mechanical properties of polypropylene. The paper contains sufficient data and analysis, and it is recommended to accept it, but several points need to be modified before considering publication.

Comments:

1.      Please unify various specialized terms in the paper and determine whether to use abbreviations or full names.

2.      The clarity of the pictures in the paper is insufficient. Please adjust the clarity and insert it again. At the same time, please adjust the table size to maintain consistency.

3.      Please confirm the formula format in the paper to ensure consistency.

 Minor editing of English language required

Author Response

Dear

Thank you for evaluating this research.

We have made all the corrections you requested.

We have also included a document with supplementary  information.

Reviewer 3 Report

The study found that the presence of furan in the polymerization process significantly impacted the productivity of the Ziegler-Natta catalyst, the MFI melt index, and the mechanical properties of polypropylene. Theoretical analysis showed that furan acts as an electron donor, while the titanium atom of the Ziegler-Natta catalyst acts as an electron acceptor, suggesting that furan may interact with the catalyst, forming a complex that blocks active sites and decreases the availability of sites for propylene polymerization. Experimental data showed that the productivity of the Ziegler-Natta catalyst decreased significantly with increasing furan concentration, and the presence of furan negatively affected the MFI and mechanical properties of polypropylene.
The paper is well written. I just have two minor comments:

line 284: please fix the format of Table 1

for all computational studied structures, please attach the optimized geometries as supplemental information.

Author Response

Dear

Thank you for evaluating this research.

We have made all the fixes you requested.

We have also included a document with supplementary information.